

# CTCN: a novel credit card fraud detection method based on Conditional Tabular Generative Adversarial Networks and Temporal Convolutional Network

Xiaoyan Zhao and Shaopeng Guan

School of Information and Electronic Engineering, Shandong Technology and Business University, Yantai, China

## ABSTRACT

Credit card fraud can lead to significant financial losses for both individuals and financial institutions. In this article, we propose a novel method called CTCN, which uses Conditional Tabular Generative Adversarial Networks (CTGAN) and temporal convolutional network (TCN) for credit card fraud detection. Our approach includes an oversampling algorithm that uses CTGAN to balance the dataset, and Neighborhood Cleaning Rule (NCL) to filter out majority class samples that overlap with the minority class. We generate synthetic minority class samples that conform to the original data distribution, resulting in a balanced dataset. We then employ TCN to analyze transaction sequences and capture long-term dependencies between data, revealing potential relationships between transaction sequences, thus achieving accurate credit card fraud detection. Experiments on three public datasets demonstrate that our proposed method outperforms current machine learning and deep learning methods, as measured by recall, F1-Score, and AUC-ROC.

## INTRODUCTION

Currently, credit card payment has become an important consumption method in modern life. However, with the rapid development of the credit card industry, fraudulent transactions have emerged as a significant problem. Credit card fraud not only results in financial losses but also damages the reputation of financial institutions, causing people to lose trust in credit card payments. Therefore, credit card fraud detection has become a crucial task for financial institutions.

Identifying credit card fraud quickly and effectively is a challenging problem. Machine learning methods have been applied in the field of credit card fraud detection to discover patterns hidden behind a large amount of data. The machine learning models used for credit card fraud detection include support vector machines (SVM), Random Forest (RF), logistic regression (LR), and Decision Tree (DT) (*Zhang et al., 2021*; *Xuan et al., 2018*; *Trivedi et al., 2020*; *Save et al., 2017*). Compared to rule-based expert knowledge systems, machine learning-based fraud detection solutions have stronger data representation capabilities

Corresponding author
Shaopeng Guan,
konexgsp@gmail.com

and can discover more fraud transaction patterns (*Arora et al., 2022*; *Alfaiz & Fati, 2022*). However, traditional machine learning methods did not consider the changing trends in consumer spending behavior, that is, they did not consider the sequential nature of credit card transaction data, leading to low detection accuracy. Fraud detection is essentially a sequential classification problem, and this characteristic is crucial for discovering more fraud transaction patterns and improving detection accuracy.

Deep Learning is a branch of machine learning that uses artificial neural networks as an architecture, which is similar to the human brain in processing data and making decisions. Deep learning has been applied to credit card fraud detection to discover the correlation between data. Currently, deep learning detection models that consider the sequential nature of transaction data include convolutional neural network (CNN), recurrent neural network (RNN), long short-term memory (LSTM), and Gate Recurrent Unit (GRU) (*Vardhani, Priyadarshini & Narasimhulu, 2019*; *Asha & KR, 2021*). These models regard transaction data as a sequential sequence, can discover the potential relationship between consumer spending behaviors, and thus improve the accuracy of fraud detection. However, credit card transactions are a type of time series and have both short-term and long-term memories. Learning both memories is essential for accurate analysis and prediction. The above deep learning models are insufficient in learning long-term memory (dependence) (*Li & Ning, 2020*; *Fan et al., 2021*). CNNs can only observe data within a historical linear size window, whereas recurrent neural networks such as LSTMs do not address the issue of gradient vanishing and gradient explosion problems.

Moreover, credit card transaction data suffers from a severe class imbalance problem, with the majority of transactions being normal data and only a small portion being fraudulent. The imbalance of credit card transaction data biases the detection results towards the majority class samples, reducing the detection performance of the model. Although there are several methods for dealing with imbalanced data, such as oversampling, undersampling, cost-sensitive learning, and ensemble learning (*Leevy et al., 2018*), they often lead to new problems. Oversampling introduces a large amount of redundant data, while undersampling is prone to losing information on normal transaction data. Cost-sensitive functions are not very applicable, and ensemble learning is susceptible to noise data (*Huang & Dai, 2021*; *Puri & Gupta, 2021*). A generative adversarial network (GAN) (*Goodfellow et al., 2014*), as a deep generation technology, can learn the distribution of complex data and generate new samples that conform to the original data distribution, improving the classification performance of imbalanced data as an oversampling algorithm. However, GAN does not consider the class overlap phenomenon in imbalanced data, which can lead to fuzzy classification boundaries and affect the performance of the detection model.

To address the issue of imbalanced data affecting fraud detection performance, we propose a credit card fraud detection method based on Conditional Tabular Generative Adversarial Networks (CTGAN) and temporal convolutional network (TCN). Firstly, we design an oversampling algorithm based on CTGAN to construct a balanced dataset. CTGAN (*Xu et al., 2019*) is a GAN-based oversampling method that can learn the distribution of complex data and generate fraudulent transaction samples that conform

to the real data distribution. To ensure the quality of the generated samples, we filter the original dataset with the Neighborhood Cleaning Rule (NCL) algorithm to remove overlapping samples in the dataset. Then, fraud transactions are efficiently and accurately detected through TCN. TCN (*Bai, Kolter & Koltun, 2018*) has a flexible receptive field, which can learn long-term dependencies by controlling the size of the receptive field, solving the problem of poor long-term dependency relationships in existing fraud detection models. Moreover, TCN allows for parallel computation, which can ensure training efficiency while learning long-term dependency relationships. Our main contributions are summarized as follows:

- We present an innovative approach for detecting credit card fraud, known as CTCN. This methodology unveils latent connections among consumer expenditure patterns, thereby enhancing the discernment precision within the underrepresented sample category. Consequently, it contributes to the overall accuracy augmentation in the identification of deceptive transactions.
- To uphold the intrinsic attributes of the primary dataset more effectively, we opt for a profound generative model as opposed to conventional oversampling techniques. This deep generative model adeptly assimilates the distribution of minority class instances, enabling the synthesis of analogous instances that faithfully mirror the authentic dataset distribution, thereby facilitating the construction of a balanced dataset.
- Within the realm of credit card datasets, a challenge emerges due to class overlapping. Employing the NCL approach to enhance CTGAN, we strategically employ the NCL algorithm to filter the foundational dataset. The objective is to excise instances entangled through overlap, concurrently ensuring the caliber of instances generated by CTGAN.

The rest of this article is organized as follows: 'Related work' introduces existing methods for imbalanced data processing and credit card fraud detection. 'Preliminary knowledge' describes the basic principles of relevant technologies. 'Proposed credit card fraud detection method' proposes our specific approach. 'Experiments and analysis' evaluates the approach through experiments and analyzes the results. Finally, 'Conclusion' summarizes the work of the entire article.

## RELATED WORK

The design of a credit card fraud detection method needs to consider two aspects (*Rtayli & Enneya, 2020*): firstly, solving the problem of class imbalance in the dataset to ensure the performance of the detection method; secondly, selecting a suitable classifier to improve the accuracy of fraud transaction detection.

Currently, methods for handling imbalanced data mainly include data-level methods and algorithm-level methods (*Das, Mullick & Zelinka, 2022*). Algorithm-level methods require a deep understanding of classification algorithms and loss functions. In this article, we focus on data-level methods. Common data-level methods include oversampling and undersampling methods. *Singh, Ranjan & Tiwari (2022)* analyzed the impact of various data-level class balancing methods on different classification algorithms, such as SMOTE, adaptive synthetic sampling (ADASYN), random oversampling (ROS), random

undersampling (RUS), Tomeklinks, cluster centroids undersampling technique, AIIKNN, SMOTE + Tomek (SMOTETomek), and SMOTE + ENN (SMOTEENN). The study showed that both oversampling and undersampling methods can solve the problem of imbalanced data. However, undersampling can lead to the loss of information in majority class samples , making it difficult for the model to fully utilize the existing information. The samples generated by oversampling lack diversity, which can cause overfitting to some extent (*Puri & Gupta, 2021*). GAN, as a new type of oversampling method, can also be used to solve the problem of imbalanced data. To this end, *Engelmann & Lessmann (2021)* proposed a Conditional Wasserstein GAN-based oversampling method for credit scoring. The results showed that this method can effectively model tabular data to solve the problem of imbalanced data. However, GAN-based oversampling methods do not consider the problem of class overlap in imbalanced data, which can cause blurry classification boundaries and thus affect the detection performance of the model.

In the past decade, machine learning and deep learning methods have been widely used in credit card fraud detection (*Carcillo et al., 2021*). *Li et al. (2021a)* established an optimal credit card fraud prediction model based on SVM by comparing and studying four different kernel functions and three parameter optimization methods. *Wang & Han (2019)* proposed a credit card fraud prediction model based on clustering analysis and ensemble SVM. They combined K-means clustering with AdaBoost ensemble to improve SVM's classification and prediction ability on imbalanced datasets. These methods can discover more fraud patterns by learning effective features from the data. However, they only analyze individual transaction information (such as amount and time) and do not consider the sequential information between consumer spending behaviors. Fraud transaction detection is essentially a sequential classification problem, and *Zhang et al. (2018)* proposed a CNN model based on feature ranking for fraud transaction detection. This method can achieve better performance by using only original features for training and saves a lot of calculation time for deriving variables. However, it does not consider long-term dependencies between transaction sequences. *Jurgovsky et al. (2018)* viewed fraud detection as a sequence classification task and used LSTM to discover hidden sequence patterns. *Forough & Momtazi (2021)* proposed an ensemble model based on data sequence modeling, using deep recurrent neural networks and artificial neural network-based voting mechanisms to detect fraud behaviors. *Benchaji et al. (2021)* applied the attention mechanism to LSTM recurrent networks. These methods view fraud detection as a sequence classification problem, which improves the performance of the detection model. However, they mainly rely on recurrent neural networks such as LSTM and GRU, which require a large amount of memory to store unit states when processing long-term sequence data (*Fan et al., 2021*), and suffer from the problem of gradient vanishing and exploding.

TCN is a novel algorithm used to solve time series prediction problems (*Lea et al., 2016*). Initially, TCN was mainly used for action segmentation in videos. *Bai, Kolter & Koltun (2018)* compared TCN with recurrent neural networks represented by LSTM/GRU and found that in processing temporal tasks, TCN is not only more accurate than classical LSTM and GRU but also has a simpler and clearer structure. In addition, *Yan et al. (2020)* applied TCN to weather forecasting, and the experimental results showed that TCN performed

better than LSTM in time series data prediction. *Chen et al. (2020)* proposed a TCN-based prediction framework to learn correlations between sequences, and the experimental results showed that this framework outperforms state-of-the-art methods in point prediction and probability prediction tasks.

In summary, to address the challenges of data imbalance processing and credit card fraud detection, we propose a credit card fraud detection method based on CTGAN and TCN.

## PRELIMINARY KNOWLEDGE

### GAN and CTGAN

Generative adversarial network are generative models based on the theory of zero-sum games. GAN consist of a generator G and a discriminator D (*Goodfellow et al., 2014*). During training, the generator continually improves its ability to create fake data to deceive the discriminator, while the discriminator judges whether the input data is real or generated. The two components iteratively optimize each other until they reach a dynamic equilibrium. The generator finally generates simulated samples and completes data augmentation. The loss function of GAN is shown in Eq. (1):

$$\min_{G}\max_{D}V(G,D) = E_{x \sim P_r}\{\log[D(x)]\} + E_{z \sim P_z}\{\log[1 - D(G(z))]\} \tag{1}$$

where $x$ represents real sampling, $P_r$ represents the real sampling distribution, $z$ represents random noise, $P_z$ represents random noise distribution, $G(z)$ represents fake sample data generated by generator $G$, and $D(\cdot)$ represents the output value of the discriminator $D$.

Although GAN can generate synthetic samples that conform to the real data distribution, they are not suitable for generating tabular data. CTGAN is a generative model based on GAN that has been optimized for the generation task of tabular data (*Xu et al., 2019*). CTGAN takes into account the conditional information in tabular data, and uses special generator and critic structures as well as other techniques. The architecture of CTGAN is shown in Fig. 1.

CTGAN consists of two neural networks: a generator G and a Critic C (similar to the discriminator in classical GAN architecture). To overcome the non-Gaussian and multimodal distribution of continuous columns in tabular data, CTGAN employs mode-specific normalization. A conditional generator and training by sample are used to address the issue of imbalanced categories in discrete columns. Additionally, CTGAN incorporates some recent advances in GAN training, such as the loss function of WGAN-GP (*Gulrajani et al., 2017*) and the critic structure of PacGAN (*Lin et al., 2018*), to improve training stability and the quality of generated data. The loss function of CTGAN is shown in Eq. (2):

$$L = E_{G(z) \sim P_g}[D(G(z))] - E_{x \sim P_r}[D(x)] + \lambda E_{y \sim P_y}[(||\nabla_y D(y)|| - 1)^2] \tag{2}$$

where $y$ represents the sample linearly interpolated to the real data $x$, $\lambda$ represents the gradient penalty factor, $P_r$ and $P_g$ represent the distribution of real and generated data.

The credit card dataset is a type of tabular data that contains both data and classification information. CTGAN is specifically designed to generate tabular data and can effectively

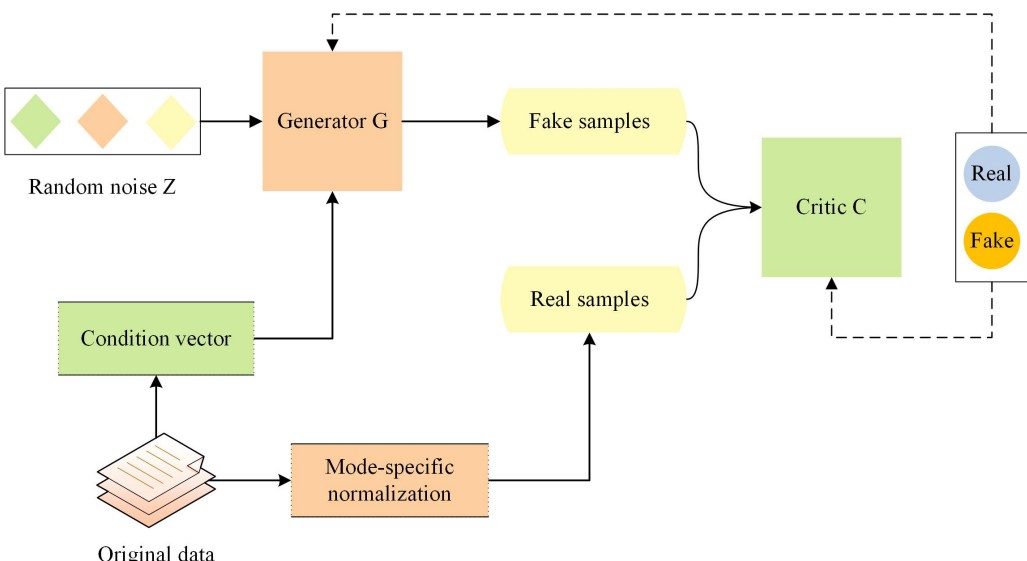

**Figure 1  Architecture of conditional tabular GAN.**

learn the distribution of credit card data, producing synthetic samples that conform to the real data distribution. This can be useful for data augmentation while maintaining data utility.

## Temporal convolutional network

Temporal Convolutional Network is a type of convolutional neural network specifically designed for processing time-series data. Its structure includes causal convolutions, dilated convolutions, and residual connections (*He & Zhao, 2019*), as illustrated in Fig. 2.

In time series processing, causal convolution ensures that the model does not use future information when processing sequential data. Dilated convolution can expand the receptive field of convolution operations and increase the model's perception range, while residual connections can speed up network convergence and improve model accuracy. These unique design features enable TCN to perform well in time series data processing.

To capture long-term historical information in time series processing, the depth of causal convolution can be increased or the convolution kernel can be enlarged (*Li et al., 2020*). However, increasing the size of the convolution kernel leads to an increase in the number of network weight parameters, and increasing the convolution depth can result in problems such as gradient vanishing, increased training complexity, and poor fitting effect. To overcome these issues, the concept of dilation was introduced into convolutional networks. Dilated convolution can increase the receptive field according to the dilation factor, enhancing the network's learning and memory capabilities over longer periods of time (*Yu & Koltun, 2015*). A dilated causal convolution with a dilation factor $d = 1,2,4$ and a convolution kernel $k = 3$ is shown in Fig. 3.

Dilated convolution allows the filter to be applied to a region larger than the length of the filter itself by skipping part of the input (*He & Zhao, 2019*). For an input sequence

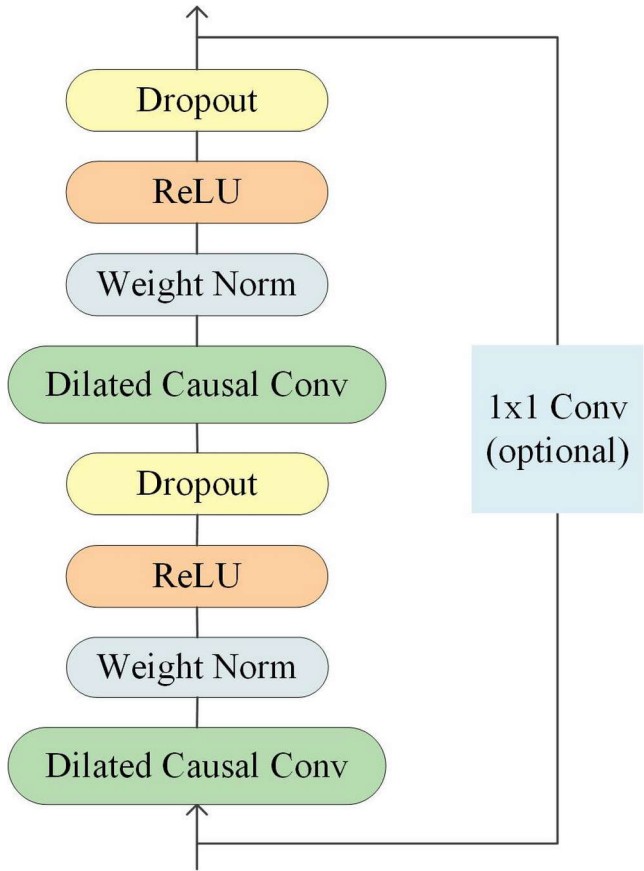

**Figure 2** **Architecture of temporal convolutional network.**

$X = (x_1, x_2, \cdots, x_T)$ of length $T$ and a filter $f : 0, \cdots, k-1$, the dilated convolution operation $F$ on element $s$ in the sequence is shown in Eq. (3).

$$F(s) = (X_{*d}f)(s) = \sum_{i=0}^{k-1} f(i) \cdot X_{s-d \cdot i} \tag{3}$$

where $d$ is the dilated factor, $*$ indicates the convolution operation, $k$ is the filter size, and $s$-$d \cdot i$ is for the direction of the past.

Since the receptive field size of TCN depends on the network depth $n$, filter size $k$, and dilated factor $d$, it is crucial to ensure stability as TCN becomes deeper and larger. To address this issue, a generic residual module can be used instead of a regular convolutional layer, which includes two layers of dilated causal convolutions and non-linear mapping. Weight normalization and dropout are also added to each layer for network regularization (*Bai, Kolter & Koltun, 2018*).

## PROPOSED CREDIT CARD FRAUD DETECTION METHOD

Detecting credit card fraud is challenging due to the imbalanced nature of the data and the need for effective fraud detection algorithms. In this section, we present CTCN, a credit

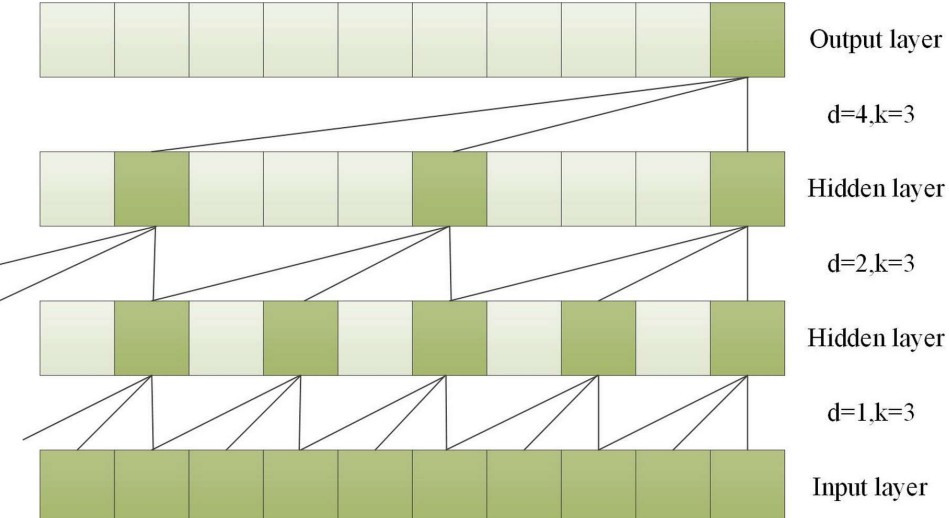

**Figure 3** A dilated causal convolution with dilated factors $d = 1,2,4$ and convolution kernel k = 3.

card fraud detection method that addresses these challenges. The system architecture of CTCN is shown in Fig. 4.

CTCN consists of three main components: data preprocessing, an improved CTGAN model, and a TCN detection algorithm. The steps involved in using CTCN to detect credit card fraud transactions are outlined below.

- **Data preprocessing**

Data preprocessing is a crucial step in building a robust and reliable fraud detection system. The data preprocessing component of CTCN includes three steps: data normalization, feature selection, and data partitioning.
1. *Data normalization*: When different features have different value ranges, the model convergence speed is slow, and it may not find the optimal value, thereby affecting the performance of the model. In order to improve the comparability of the data, the model uses *Z*-Score normalization (*Jain, Shukla & Wadhvani, 2018*) to normalize the data to the $[-1,1]$ interval.
2. *Feature selection*: Too many features will increase the complexity of the model, leading to overfitting, while too few features will result in insufficient fitting of the model. The model uses gradient boosting decision tree (GBDT) feature importance (*Ji et al., 2021*) for feature selection to select appropriate input features.
3. *Data set partitioning*: The original data set is divided into a training set and a test set. The training set is used to train the improved CTGAN and TCN detection algorithms, while the test set is used to test the generalization ability of the model.

- **Improvement of CTGAN model**

Data imbalance can affect the performance of detection models, and traditional oversampling methods cannot effectively learn the sample distribution, resulting in

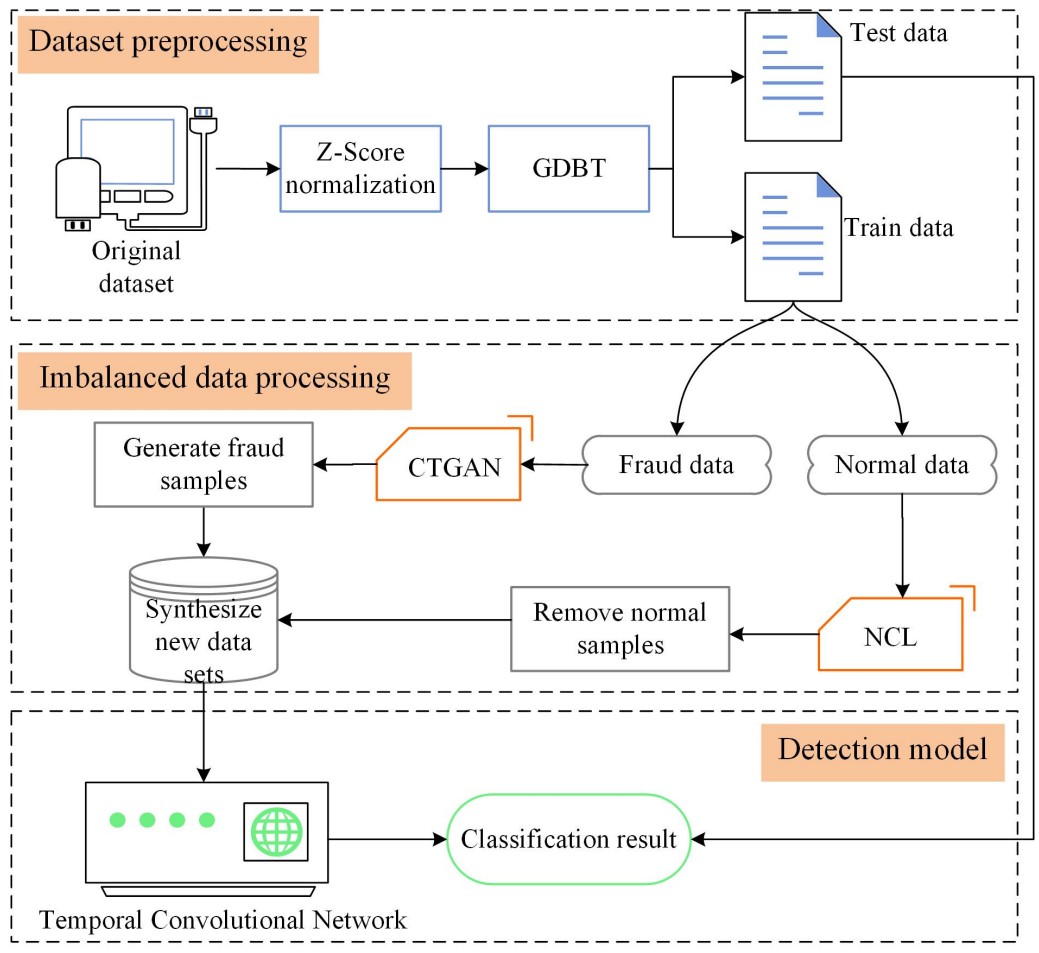

**Figure 4   CTCN System Architecture.**

generated data that is not realistic. We use the deep generative model CTGAN instead of traditional oversampling methods to generate fraud transaction samples that conform to the real data distribution. The oversampling algorithm based on CTGAN can generate data based on the distribution of the real data set, effectively solving the problem of data imbalance in the credit card data set. However, in the credit card data set, normal transaction data contains noisy and redundant samples, and overlaps with minority class samples, which affects the quality of the generated data by CTGAN. In the fraud detection field, minority class samples in overlapping areas are more representative. In order to improve the quality of the generated minority class samples, avoid generating overlapping and noisy samples, we use the NCL algorithm (*Laurikkala, 2001*) to improve the CTGAN model and design an imbalanced data processing algorithm based on the improved CTGAN. The NCL algorithm removes majority class samples in overlapping regions through clustering, and then CTGAN is used to construct a balanced data set. The improved CTGAN adeptly generates top-tier minority class samples that adhere faithfully to the authentic data distribution. It effectively rectifies the imbalance conundrum within

the source dataset, thereby orchestrating a heightened focus of the detection algorithm on the intricacies and patterns exhibited by the minority class instances. By ameliorating the discernment accuracy of these specific instances, it consequently amplifies the holistic precision of the detection algorithm.

- **TCN detection algorithm**

The TCN detection algorithm is used to analyze the transaction sequence to capture the long-term dependencies between data and discover potential correlations among consumer spending behaviors to detect fraudulent transactions. Firstly, an unbalanced dataset is used to train TCN and determine the network hyper-parameters. Then, the model is used to detect fraudulent transactions.

The steps of the improved CTGAN algorithm are outlined in Algorithm 1.

---

**Algorithm 1** Improved CTGAN Algorithm

---

**Require:** Original training set $X$, Number of nearest neighbors: $N$

**Ensure:** New balanced training set $X'$

1: Divide the original training set $X$ into normal transaction samples $X_n$ and fraudulent transaction samples $X_f$ , and initialize $R$ as an empty set

2: **for** each sample $x \in X_f$ **do**

3:     $y$ is samples other than $x$, $Y = X - x$

4:     **for** $y \in X$ **do**

5:     $$d(x,y) = \sqrt{\sum_{i=1}^{n} (x_i - y_i)^2}$$

6:     **end for**

7:     Sort the distance values between $x$ and all $y$ in increasing order

8:     Find the first $N$ nearest neighbors $N_x$ of $x$, $N_x = \{x_1, x_2, \ldots, x_N\}$

9:     **for** $x_j \in N_x$ **do**

10:         **if** $x_j \in X_n$ **then**

11:             $R = R \cup x_j$

12:         **end if**

13:     **end for**

14: **end for**

15: Remove the majority class samples that belong to the set $R$ from $X_n$ to obtain the filtered training set $X'_n$: $X'_n = X_n - R$

16: Train the CTGAN model using the fraudulent transaction dataset $X_f$ and iterate the generator $G$ and critic $C$

17: Use the trained CTGAN model to generate the specified number of fraudulent transaction samples $X_{new}$ and add them to $X_f$: $X'_f = X_{new} \cup X_f$

18: Combine $X'_n$ and $X'_f$ to obtain the new balanced training set $X'$: $X' = X'_n \cup X'_f$

---

In the algorithm, the initial step involves dividing the original training set into majority class dataset (normal transaction dataset) and minority class dataset (fraudulent transaction dataset), and creating an empty set $R$ (line 1). Subsequently, the NCL is utilized to filter

**Table 1  Credit card data set description.**

| Dataset | Total transactions | Features | Normal | Fraudulent | Fraud/Total ratio (%) |
|---------|-------------------|----------|--------|------------|----------------------|
| Europe | 284,807 | 31 | 284,315 | 492 | 0.172 |
| Taiwan | 30,000 | 25 | 23,364 | 6,636 | 22.12 |
| German | 1,000 | 21 | 700 | 300 | 30.00 |

the original dataset and eliminate overlapping and noisy samples. For each minority class sample, $x$, its Euclidean distance is calculated from other samples $y$ in the training set (lines 2–6). The first $N$ nearest neighbors $N_x$ of $x$ are then identified based on the Euclidean distance (lines 7–8). If any of the $N$ samples $x_j$ belongs to the majority class dataset, $x_j$ is added to the set $R$ (lines 9–14). The majority class samples in the set $R$ are then removed from $X'_n$ to generate the filtered dataset (line 15). Next, the minority class dataset is employed to train the CTGAN model, which can better learn the minority class distribution (line 16). Using the generative network of CTGAN, the specified number of fraudulent transaction samples $X_{new}$ are generated (line 17). Finally, the filtered majority class dataset is mixed with the extended minority class dataset to obtain the new balanced training set (line 18).

Collectively, by means of meticulous data preprocessing, the dataset stands poised for optimization, culminating in an ameliorated efficacy of the detection algorithm. This refinement serves a dual purpose: to curtail the peril of overfitting and to tailor the dataset more aptly to the nuances of the detection algorithm. The enhancement of CTGAN yields a concomitant augmentation in the algorithm's capacity to discern disparate categories, concurrently mitigating the toll of misclassification. This, in turn, engenders an overarching enhancement in the algorithm's comprehensive performance. Paired with the TCN detection algorithm, it unveils latent correlations inherent in consumer expenditure behavior. This, in turn, conduces to the unearthing of concealed fraudulent patterns, thereby ushering in a heightened elevation of the algorithm's collective detection prowess. Synthesized harmoniously, these merits synergistically underpin substantial strides in the algorithm's efficacy, precision, and applicative value.

## EXPERIMENTS AND ANALYSIS

We conducted comparative experiments between our improved CTGAN algorithm and other oversampling, undersampling, and hybrid sampling methods to evaluate its performance in addressing the imbalance in transaction data. Moreover, we compared CTCN with traditional machine learning and deep learning methods to test the detection effectiveness of the proposed method. The experiments were conducted on a computer system with an Intel Core i5 10400F 2.90 GHz CPU and 8.0GB RAM.

### Experimental dataset

We have used three different datasets from the real world to evaluate the proposed method. These datasets exhibit dissimilar transaction volumes and proportions of deceptive transactions, with the specificities expounded in Table 1.

**Table 2  Credit card fraud transaction detection confusion matrix.**

|  | Predicted normal | Predicted fraudulent |
| --- | --- | --- |
| Actual normal | True Positive($TP$) | False Negative($FN$) |
| Actual fraudulent | False Positive($FP$) | True Negative($TN$) |

The Europe dataset is a real credit card transaction dataset used in *Dal Pozzolo et al. (2015)*, comprising 284,807 transaction records spanning two days of European credit card holders. The dataset includes 31 feature vectors, such as time, transaction amount, other attributes (V1 to V28) processed *via* PCA, and a "Class" attribute distinguishing fraudulent and normal transactions. The "Class" attribute takes a value of 0 for normal transactions and 1 for fraudulent transactions, with 492 fraud transactions present in the dataset.

The Taiwan dataset (*Yeh & Lien, 2009*) is a credit card user payment dataset obtained from the UCI machine learning repository. It includes 30,000 transaction records, of which 6,636 are default records. The dataset comprises 25 feature vectors, such as credit card user's overdue payment, demographic factors, credit data, payment history, and bills. The "default.payment.next.month" attribute indicates whether the credit card is overdue and takes a value of 0 or 1.

The German dataset is the South German credit data released on the UCI database, consisting of 1,000 transaction records, of which 300 belong to customers with poor credit. The dataset contains 21 features representing customer financial status, such as financial record status, measures of prepayments, bank accounts or securities, business terms, installment payment rates of additional cash levels, property, age, and the number of existing credits.

## Evaluation metrics

We used a confusion matrix to evaluate the performance of our model in detecting credit card fraud. The confusion matrix, shown in Table 2, is used to calculate several evaluation metrics.

In this matrix, $TP$ indicates the number of actual normal transactions and predicted to be normal transactions, $FP$ indicates the number of transactions that are actually fraudulent but are predicted to be normal, $FN$ indicates the number of transactions that are actually normal but are predicted to be fraudulent, $TN$ indicates actual fraudulent transactions and predicts the number of fraudulent transactions.

Using the parameters from the confusion matrix, we define the following evaluation metrics (*Li et al., 2021b*):

Accuracy is the probability of correctly classified transactions in all transactions and is defined as:

$$Accuracy = \frac{TP + TN}{TP + TN + FP + FN} \tag{4}$$

Recall is the probability of being predicted as a normal transaction in the sample of normal transactions and is defined as:

$$Recall = \frac{TP}{TP + FN} \tag{5}$$

Precision is the probability that a normal transaction in the sample of normal transactions is actually a normal transaction and is defined as:

$$Precision = \frac{TP}{TP + FP} \tag{6}$$

F-Score is a metric that combines precision and recall and is defined as:

$$F - \text{Score} = (1 + \beta^2) \cdot \frac{Precision \cdot Recall}{\beta^2 \cdot Precision + Recall} \tag{7}$$

In Eq. (7), $\beta$ is the coefficient that describes the relative importance of precision and recall. When $\beta = 1$, the importance of precision and the importance of recall are equivalent, and we obtain the F1-Score:

$$F1 - \text{Score} = \frac{2 \cdot Precision \cdot Recall}{Precision + Recall} \tag{8}$$

The AUC-ROC measures the generalization ability of the classification problem model. The ROC curve consists of two parameters: the TPR and the FPR, which are defined as:

$$TPR = \frac{TP}{TP + FN} \tag{9}$$

$$FPR = \frac{FP}{TN + FP} \tag{10}$$

For different classification thresholds, we can obtain a series of TPR and FPR values to plot the ROC curve. The AUC is a measure of the area under the ROC curve from (0,0) to (1,1), which is one of the important metrics for performance evaluation of classification problems (*Forough & Momtazi, 2022*).

The Matthews Correlation Coefficient (MCC) is an important measure of classification performance for binary classification problems and is defined as:

$$MCC = \frac{TP \cdot TN - FP \cdot FN}{\sqrt{(TP + FP)(TP + FN)(TN + FP)(TN + FN)}} \tag{11}$$

This metric can also be used to assess the classification performance of a binary classification problem when the samples are extremely unbalanced (*Chicco, Tötsch & Jurman, 2021*). Specificity is the probability that a sample of fraudulent transactions will be correctly predicted as fraudulent transactions:

$$Specificity = \frac{TN}{(FP + TN)} \tag{12}$$

The combined normal transaction recall and fraudulent transaction recall metric, G-mean, is defined as follows:

$$G - \text{mean} = \sqrt{Recall \times Specificity} = \sqrt{\frac{TP}{TP + FN} \times \frac{TN}{TN + FP}} \tag{13}$$

This metric is useful for evaluating the performance of a classification model when the data is imbalanced (*Aurelio et al., 2019*).

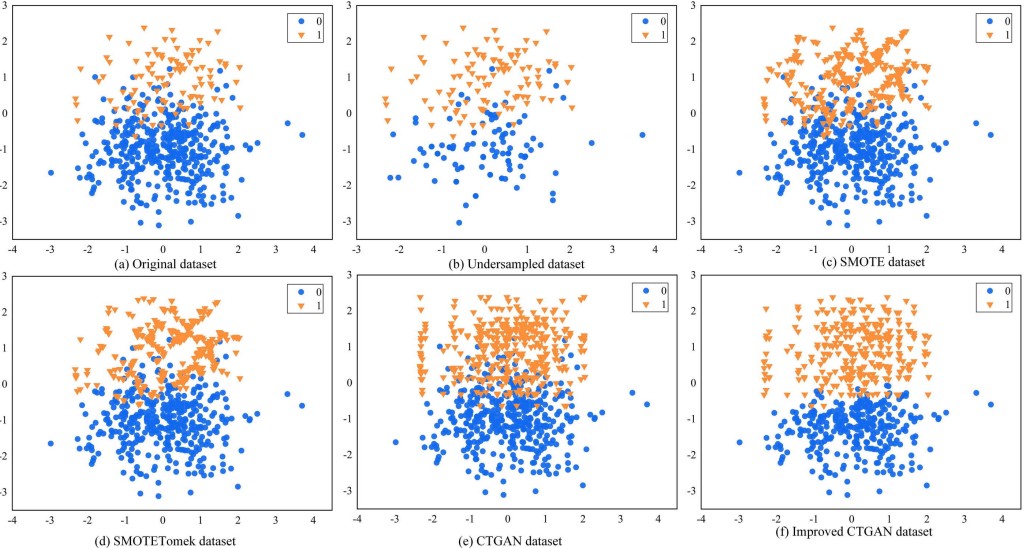

**Figure 5**  Different sampling methods sampling effect comparison graph.

## Experimental results

### Performance experiment of improved CTGAN

To evaluate the performance of the improved CTGAN in handling imbalanced data, we randomly generated a set of artificially imbalanced datasets using the Python 3.6 sklearn package, which consisted of 500 data points. Among them, the majority class had 399 samples, and the minority class had 101 samples. We then conducted experiments with undersampling RUS, oversampling SMOTE, hybrid sampling SMOTETomek, CTGAN, and improved CTGAN. Figure 5 shows the visualization results of these sampling methods on the dataset.

Figure 5A illustrates the distribution of the original data. We observe class overlap between minority and majority class samples, which leads to a blurred classification boundary. Figure 5B shows the data distribution after RUS processing, which randomly removes majority class samples to make the number of majority and minority class samples equal. However, this method may result in the loss of some useful information, and the class overlap still persists in the dataset. Figure 5C shows the data distribution after SMOTE processing, which generates new samples randomly along the lines connecting adjacent minority class samples. However, the generated new samples are similar to the original samples, and some of them fall into the majority class region, further exacerbating class overlap. Figure 5D shows the data distribution after SMOTETomek processing, which employs the Tomek link algorithm for data cleaning based on SMOTE, removing redundant data. We observe that SMOTETomek can eliminate redundant data, but the class overlap and blurred classification boundary persist. Figures 5E and 5F show the data distribution after CTGAN and improved CTGAN processing, respectively. Both methods generate new samples that are distributed similarly to the real samples. However, some of the new samples generated by CTGAN fall into the majority class region, exacerbating

**Table 3** F1-Score, MCC and G-mean based on different sampling methods.

| Methods | Europe | | | Taiwan | | | German | | |
|---|---|---|---|---|---|---|---|---|---|
| | F1-Score | MCC | G-mean | F1-Score | MCC | G-mean | F1-Score | MCC | G-mean |
| Original | 0.8026 | 0.8040 | 0.8677 | 0.4675 | 0.3886 | 0.5892 | 0.4533 | 0.3298 | 0.5691 |
| RUS | 0.0283 | 0.1082 | 0.9107 | 0.5250 | 0.3961 | 0.6738 | 0.5913 | 0.3942 | 0.7131 |
| ROS | 0.7034 | 0.7108 | 0.9090 | 0.4785 | 0.3030 | 0.6803 | 0.5784 | 0.3921 | 0.6970 |
| NN | 0.0037 | 0.0109 | 0.3456 | 0.3846 | 0.1408 | 0.5786 | 0.5690 | 0.3554 | 0.6937 |
| NCL | 0.8152 | 0.8153 | 0.8887 | 0.4955 | 0.3335 | 0.6819 | 0.5543 | 0.3239 | 0.6603 |
| SMOTE | 0.6302 | 0.6475 | 0.9066 | 0.5314 | 0.4030 | 0.6800 | 0.5748 | 0.3610 | 0.6958 |
| ADASYN | 0.5576 | 0.5881 | 0.9063 | 0.4700 | 0.2899 | 0.6738 | 0.5618 | 0.3420 | 0.6557 |
| SMOTEENN | 0.5520 | 0.5849 | 0.9100 | 0.4783 | 0.3031 | 0.6815 | 0.5530 | 0.3297 | 0.6575 |
| SMOTETomek | 0.7150 | 0.7205 | 0.9057 | 0.4955 | 0.3335 | 0.6819 | 0.5859 | 0.3798 | 0.7053 |
| CTGAN | 0.8037 | 0.8034 | 0.8922 | 0.5134 | 0.3881 | 0.6588 | 0.6042 | 0.4130 | 0.7243 |
| Improved CTGAN | **0.8187** | **0.8185** | **0.9108** | **0.5349** | **0.4060** | **0.6841** | **0.6108** | **0.4227** | **0.7300** |

**Notes.**

The values in bold indicate the best results.

the overlap between minority and majority class samples. In contrast, improved CTGAN generates samples that conform to the real data distribution in the distribution area of the original samples, without generating noisy samples.

To further validate the performance of the improved CTGAN in handling imbalanced data, we conducted an experiment using TCN as the fraud detection algorithm to detect the dataset processed by CTGAN. We then compared the results with those of the other 10 credit card datasets. These 10 datasets included the original imbalanced dataset and nine balanced datasets processed by ROS, RUS, SMOTE, ADASYN, Near Miss(NN), NCL, SMOTEENN, SMOTETomek, and CTGAN sampling methods. We used F1-Score, MCC, and G-mean as evaluation metrics. F1-Score measures the accuracy and recall of the model, MCC measures the classification performance of the model, and G-mean measures the data imbalance. The experimental results are presented in Table 3.

The experimental results presented in Table 3 demonstrate the superior performance of the improved CTGAN in handling imbalanced data. In particular, the datasets processed by the improved CTGAN achieved the best results in all three metrics of F1-Score, MCC, and G-mean, with significant improvements observed in the German dataset, where the metrics were improved by 15%, 9%, and 16%, respectively. This can be attributed to the fact that the improved CTGAN filters the original data by removing most of the overlapping majority class samples and generates minority class samples that conform to the true data distribution to construct a balanced dataset. It is worth noting that while CTGAN also performed well in terms of F1-Score and G-mean metrics, the newly generated samples by CTGAN tended to fall into the majority class region, exacerbating the overlap between majority and minority class samples and making the classification boundary between them blurred.

RUS, NN, and NCL construct a balanced dataset by removing majority class samples, and the three datasets processed by these three undersampling algorithms achieved good experimental results in the G-mean metric. However, undersampling algorithms can lead

**Table 4** Accuracy, Recall and F1-Score based on different detection algorithms.

| Methods | Europe | | | Taiwan | | | German | | |
|---|---|---|---|---|---|---|---|---|---|
| | Accuracy | Recall | F1-Score | Accuracy | Recall | F1-Score | Accuracy | Recall | F1-Score |
| DT | **0.9993** | 0.7839 | 0.8141 | 0.8250 | 0.3480 | 0.4617 | 0.7454 | 0.3229 | 0.4246 |
| LR | 0.9991 | 0.5925 | 0.7058 | 0.8226 | 0.3185 | 0.4364 | **0.7909** | 0.5626 | 0.6101 |
| SVM | 0.9992 | 0.6111 | 0.7415 | 0.8233 | 0.3634 | 0.4701 | 0.7666 | 0.4895 | 0.5497 |
| RF | **0.9993** | 0.7160 | 0.7972 | 0.8235 | 0.3367 | 0.4514 | 0.7424 | 0.2291 | 0.3410 |
| CNN | **0.9993** | 0.7551 | 0.8000 | 0.8223 | 0.3578 | 0.4648 | 0.7363 | 0.6875 | 6027 |
| ANN | – | 0.7416 | 0.7648 | 0.8229 | 0.3854 | 0.4842 | 0.7696 | 0.4895 | 0.5529 |
| LSTM | – | 0.7408 | 0.7866 | 0.8238 | 0.3704 | 0.4756 | 0.7787 | 0.4687 | 0.5521 |
| GRU | – | 0.7208 | 0.7792 | 0.8250 | 0.3817 | 0.4848 | 0.7515 | 0.4895 | 0.5340 |
| LSTM-Attention | **0.9993** | 0.7901 | 0.8101 | 0.8232 | 0.3592 | 0.4671 | 0.7757 | 0.4687 | 0.5487 |
| RNN-LSTM | **0.9993** | 0.7777 | 0.8000 | 0.8224 | 0.3770 | 0.4780 | 0.7818 | 0.5208 | 0.5813 |
| CNN-GRU | **0.9993** | 0.7716 | 0.8143 | **0.8254** | 0.3629 | 0.4728 | **0.7909** | 0.5625 | 0.6101 |
| CTCN | **0.9993** | **0.8299** | **0.8187** | 0.7981 | **0.5381** | **0.5349** | 0.7181 | **0.7604** | **0.6108** |

**Notes.**
The values in bold indicate the best results.

to the loss of some useful samples in the majority class, which can affect the detection results of the detection algorithm. For example, the F1-Score and MCC metrics of the Europe dataset after being processed by RUS were only 0.0283 and 0.1082, respectively. ROS, SMOTE, ADASYN, SMOTEENN, and SMOTETomek are five sampling algorithms that construct a balanced dataset by expanding the minority class samples. The experimental results of their processed data in F1-Score, MCC, and G-mean metrics were also better than those of the original dataset. However, these five sampling algorithms only start from the local neighborhood of the minority class samples and do not consider the overall distribution of the minority class samples. The generated data cannot effectively fit the true data distribution and can reduce the performance of the detection algorithm. For example, the F1-Score and MCC metrics of the Europe dataset after being processed by ADASYN were 0.5576 and 0.5881, respectively.

### CTCN experiment

For the purpose of appraising the potency of our advanced methodology, we undertook a sequence of comparative experiments juxtaposing CTCN against other prevalent models employed in fraud detection, including but not limited to DT, LR, SVM, RF, CNN, ANN, GRU, LSTM, LSTM-Attention, RNN-LSTM, and CNN-GRU as detailed in references (*Forough & Momtazi, 2021*; *Forough & Momtazi, 2022*; *Roseline et al., 2022*; *Karthika & Senthilselvi, 2023*). We performed these experiments on three distinct datasets and used accuracy, recall, and F1-Score as the evaluation metrics. The results are presented in Table 4.

Based on the results presented in Table 4, our method achieved the highest accuracy on the Europe dataset, which is comparable to the performance of DT, RF, CNN, LSTM-Attention, RNN-LSTM, and CNN-GRU models. However, our method performed worse than the CNN-GRU model on the Taiwan dataset, and was inferior to the LR and CNN-GRU models on the German dataset. Given the imbalanced nature of credit card datasets,
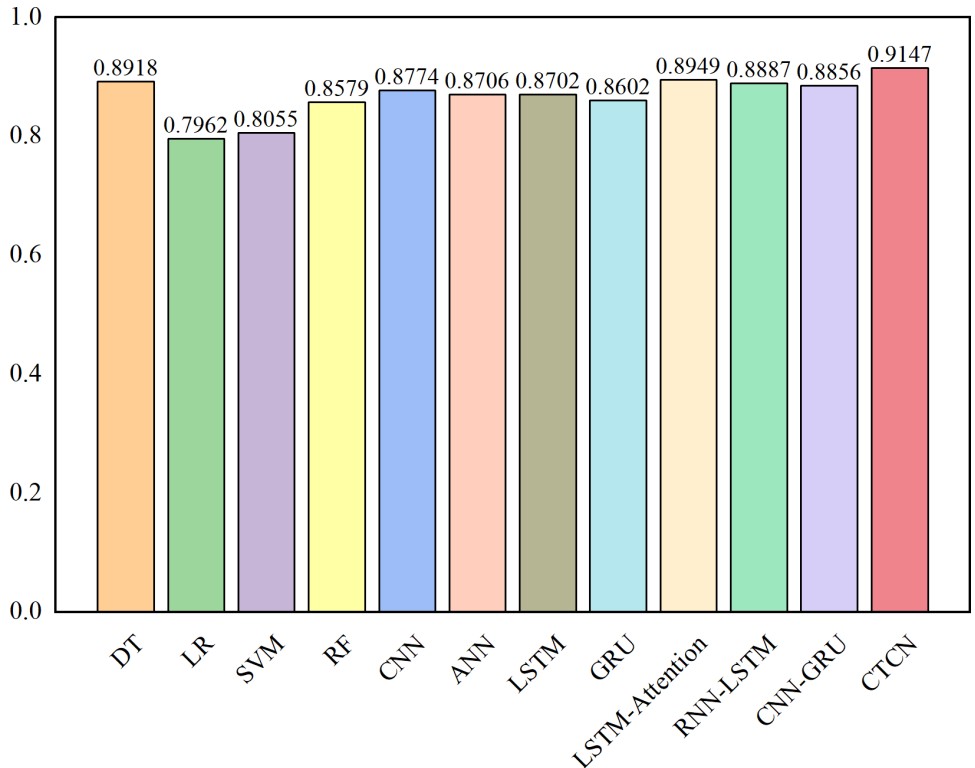

**Figure 6** AUC comparison results for the Europe dataset.

even if all transactions are classified as normal, the detection model may still achieve high accuracy. Therefore, accuracy alone is not the most informative indicator when evaluating imbalanced data classification. Instead, recall, which measures the proportion of actual normal transactions that are correctly identified as such, is a crucial metric for evaluating credit card fraud detection methods. Our method achieved the highest recall on all three datasets, indicating its superior coverage of minority classes. Additionally, the F1-Score, which combines both precision and recall, is a comprehensive evaluation index. Our method outperformed all comparison models in terms of F1-Score.

The AUC metric serves as an inherently intuitive means to assess the efficacy of the detection algorithms. We present the AUC comparative results between CTCN and eleven distinct detection algorithms, visually depicted in Figs. 6, 7 and 8.

As depicted in Figs. 6, 7 and 8, the AUC value of CTCN surpasses that of other machine learning and deep learning methods. These findings indicate that CTCN can significantly enhance the overall classification performance of the data.

To further assess the efficacy of CTCN, we have also generated ROC curves for various algorithms on the Taiwan dataset and German dataset, as shown in Fig. 9. For the Europe dataset, we employed the same evaluation metrics as those employed in *Forough & Momtazi (2022)* to make direct comparisons of experimental results between CTCN and the three

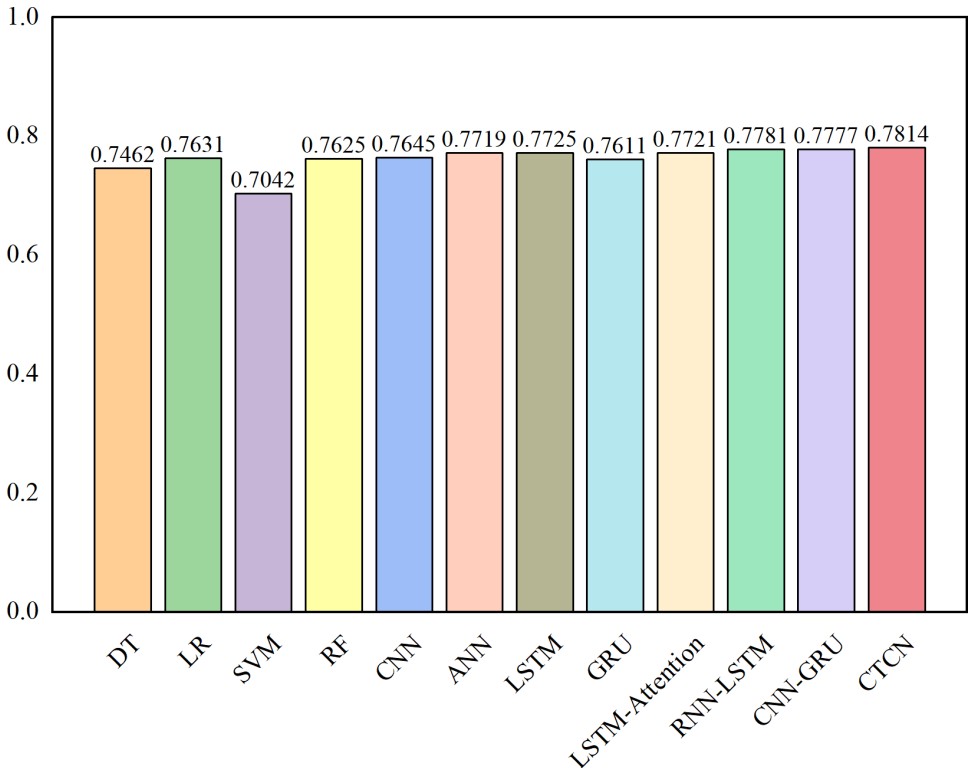

**Figure 7  AUC comparison results for the Taiwan dataset.**

algorithms, namely, ANN, GRU and LSTM in *Forough & Momtazi (2022)*. Consequently, we do not provide its ROC curves here.

Figure 9A shows the ROC curves of different algorithms on Taiwan dataset. It can be seen that SVM has the smallest ROC curve coverage area, while CTCN has the largest ROC curve coverage area, and the remaining methods have similar ROC curve coverage areas. Figure 9B shows the ROC curves of different algorithms on the German dataset. It can be seen that the ROC curve coverage area of CTCN is superior to that of the other detection algorithms. The ROC curve coverage area of CNN-GRU is marginally inferior to that of CTCN, while the ROC curve coverage area of DT is the smallest. The larger the ROC curve coverage area, the better the classification performance of the algorithm. CTCN achieved the largest ROC curve coverage area for both the Taiwan dataset and German dataset. Thus, the classification performance of CTCN is superior. In conclusion, our proposed approach outperforms popular techniques such as DT, LR, SVM, RF, CNN, ANN, GRU, LSTM, LSTM-Attention, RNN-LSTM, and CNN-GRU in credit card fraud detection tasks. It should be noted that CTCN runs slightly slower than other fraudulent transaction detection methods due to its equalization of the dataset by improving CTGAN, but the overall detection accuracy of CTCN is better than other fraudulent transaction methods.

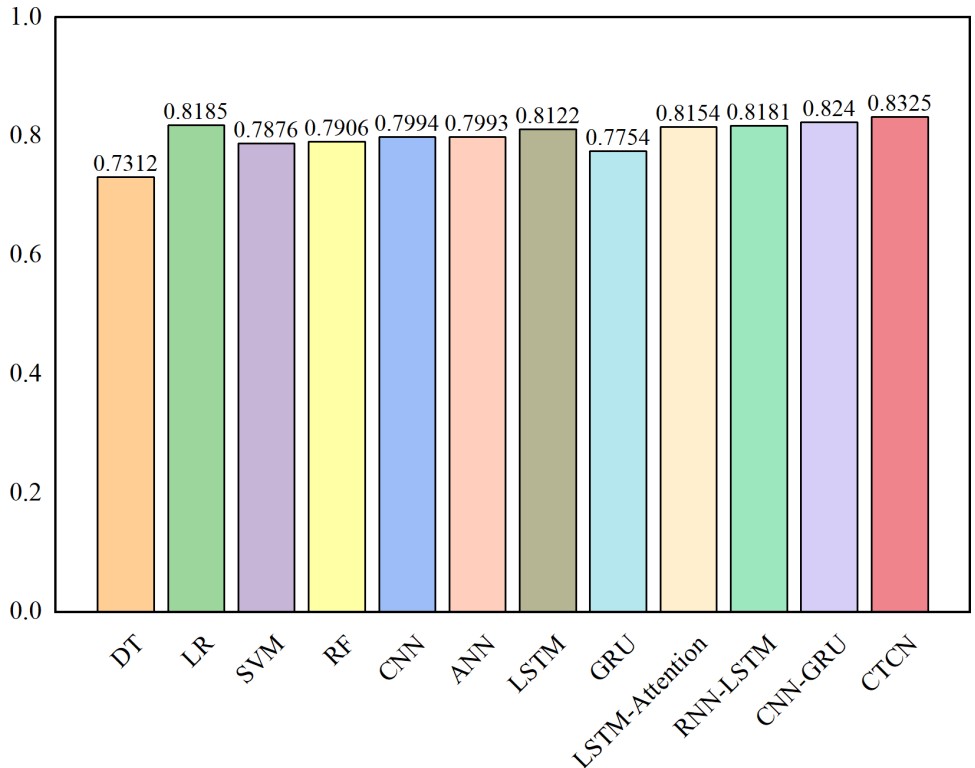

**Figure 8** AUC comparison results for the German dataset.

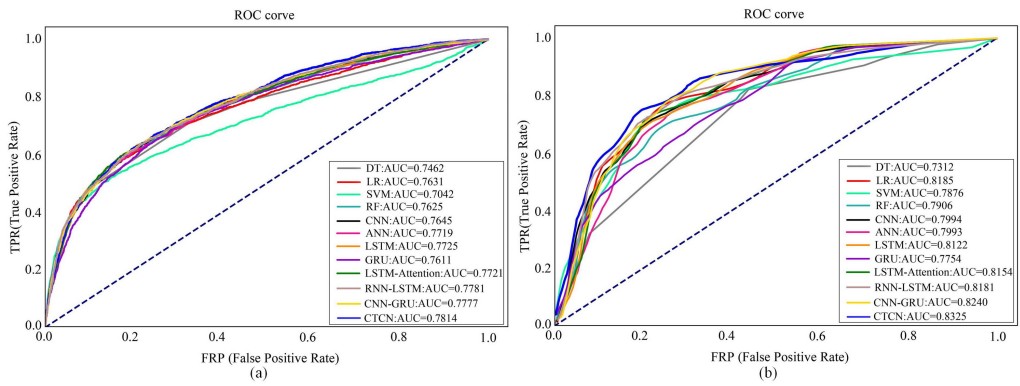

**Figure 9** ROC curves of different algorithms on different dataset. (A) Taiwan. (B) German.

## Ablation study

To delve further into the intricacies of the CTCN model, a comprehensive ablation study was undertaken, aimed at dissecting the efficacy of each constituent module. Leveraging the European dataset as the experimental foundation, the specifics of this ablation inquiry are meticulously documented in Table 5.

**Table 5  Results of the ablation study.**

| Methods | Model | | | Recall | F1-Score | AUC | G-mean |
|---|---|---|---|---|---|---|---|
| | GBDT | NCL | CTGAN | | | | |
| TCN(O) | – | – | – | 0.7407 | 0.7717 | 0.8702 | 0.8605 |
| GBDT(O) | ✓ | – | – | 0.7469 | 0.8013 | 0.8733 | 0.8641 |
| CTGAN(O) | – | – | ✓ | 0.7839 | 0.8015 | 0.8918 | 0.8852 |
| OS(N) | – | ✓ | ✓ | 0.8024 | 0.8049 | 0.9010 | 0.8956 |
| CTCN | ✓ | ✓ | ✓ | **0.8299** | **0.8187** | **0.9147** | **0.9108** |

Notes.
The values in bold indicate the best results.
TCN(O) denotes that the original data is detected using TCN; GBDT(O) denotes that the original data is feature-selected and then detected using TCN; CTGAN(O) denotes that CTGAN balances the original dataset and then detects it using TCN; OS(N) denotes that the original dataset is balanced using improved CTGAN and then TCN is used for detection; CTGAN indicates that the full module was used.

In Table 5, TCN(O) denotes the detection results of TCN based on the original dataset. GBDT(O) denotes the result of using TCN detection after feature selection on the original data. The detection results of GBDT(O) outperform those of TCN(O) due to the fact that GBDT removes irrelevant or redundant features and reduces the effect of noise. CTGAN(O) denotes the detection results on TCN using only the CTGAN oversampling processed data. The detection results of CTGAN(O) are better than those of TCN(O), CTGAN balances the original data, which makes TCN pay more attention to the features and patterns of the minority class samples and improves the recognition accuracy of the minority class samples. OS(N) denotes the detection result of the data on TCN after processing using improved CTGAN oversampling. CTGAN exacerbates the overlapping phenomenon in the original data when generating minority class samples, and from the detection results of OS(N), improved CTGAN can remove the majority class samples in the overlapping region and improve the quality of the CTGAN-generated samples in order to improve the detection performance of TCN. In summary, the modules in CTCN can effectively improve the classification performance of the classifier.

## CONCLUSION

In this study, we presented CTCN, a credit card fraud detection method that combines CTGAN and TCN to address the issue of imbalanced data. We improved CTGAN by introducing NCL to solve the problem of class overlap in imbalanced datasets, and generated minority class samples that conform to the true data distribution. This helped us to construct a balanced dataset that could be used to train our detection model. We then used TCN to analyze transaction sequences and identify potential correlations between transaction data, capturing the long-term dependency relationships between transactions. Our experiments demonstrated that the improved CTGAN outperformed the other nine sampling methods in terms of F1-Score, MCC, and G-mean metrics. Furthermore, we evaluated the proposed method on fraud detection in three different credit card datasets and compared it with popular detection models such as DT, LR, SVM, RF, CNN, ANN, GRU, LSTM, LSTM-Attention, RNN-LSTM, and CNN-GRU. The results showed that

our method achieved superior performance in terms of Recall, F1-Score, and AUC-ROC metrics.

### Funding
The authors received no funding for this work.

### Competing Interests
The authors declare there are no competing interests.

### Author Contributions
- Xiaoyan Zhao conceived and designed the experiments, performed the experiments, analyzed the data, performed the computation work, prepared figures and/or tables, and approved the final draft.
- Shaopeng Guan conceived and designed the experiments, analyzed the data, performed the computation work, authored or reviewed drafts of the article, and approved the final draft.

### Data Availability
The code is available at Zenodo: Xiaoyan Zhao. (2023). code [Data set]. Zenodo. https://doi.org/10.5281/zenodo.8161815

The third-party data is available at:

- Kaggle, https://www.kaggle.com/jacklizhi/creditcard, creditcardfraud.csv

- Kaggle, https://www.kaggle.com/uciml/default-of-credit-card-clients-dataset, UCI_Credit_Card.csv

- Kaggle, https://www.kaggle.com/datasets/agsam23/german-credit Credit_Card_Applications, german_cerdit_data.csv

- Kaggle, https://www.kaggle.com/datasets/ashkanforootan/credit-card-applications, Credit_Card_Applications.

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
