# Peer review of "CTCN: a novel credit card fraud detection method based on Conditional Tabular Generative Adversarial Networks and Temporal Convolutional Network"

_PeerJ Computer Science, doi:10.7717/peerj-cs.1634_

## Round 0.1 · original submission · Major Revisions

I have received reviews of your manuscript from scholars who are experts on the cited topic. They find the topic very interesting; however, several concerns must be addressed regarding experimental results and contributions. These issues require a major revision. Please refer to the reviewers’ comments listed at the end of this letter, and you will see that they are advising that you revise your manuscript. If you are prepared to undertake the work required, I would be pleased to reconsider my decision. Please submit a list of changes or a rebuttal against each point that is being raised when you submit your revised manuscript.

Thank you for considering PeerJ Computer Science for the publication of your research.

With kind regards,

Reviewer 1 ·

Basic reporting

No comments

Experimental design

Provide the Experimental findings clearly.

Validity of the findings

No Comments

Additional comments

CTCN: A novel credit card fraud detection method based on
CTGAN and TCN is the title of the article. credit card fraud detection is one of the most important thing nowadays.Credit card fraud can lead to signiûcant ûnancial losses for both individuals and ûnancial
institutions. A novel method called CTCN, which uses Conditional
Tabular Generative Adversarial Networks (CTGAN) and Temporal Convolutional Network
(TCN) for credit card fraud detection. Our approach includes an oversampling algorithm
that uses CTGAN to balance the dataset, and Neighborhood Cleaning Rule (NCL) to ûlter
out majority class samples that overlap with the minority class.The results showed that our method
achieved superior performance in terms of recall, F1-Score, and AUC-ROC metrics. Although, some comments for the improvement of the manuscript. Author need to address the following suggestion.
1.The abstract section is good. Although provide the experimental findings in it.
2. The main contribution in not good. Include the main contribution hierarchical way.
3. what are the advantages of data processing methods.
4. The propose algorithm is written well.
5. Check for the typos correction.
6. Include ablation study of the methods. Each feature is necessary.
7. Compare the results withe the SOTA methods.
8. Include the computational complexity matrix of all methods. Show how the proposed method is faster.

Reviewer 2 ·

Basic reporting

It very good paper. Narration of the paper need to improve.
In figure add your method name.

Experimental design

Very good, research is very good for class imbalance.

Validity of the findings

Findings are very good.

Additional comments

Improve the narration of the paper

Reviewer 3 ·

Basic reporting

• The basic reporting of the manuscript is clear. The English language used is clear and the manuscript is quite easy to follow. The structure of the paper conforms to PeerJ standards.Figures are labeled correctly and in high quality.

Experimental design

• The experimental setup seems good. The proposed method is not compared with the state-of-the-art/related works in terms of experimentation,results, and suitability.

Validity of the findings

• It looks that some significant findings were observed. However, the findings can be evaluated after clarifying the models.

Additional comments

In my view, the manuscript requires a major revision before it can be published in a journal.

---

## Round 0.2 · accepted · Accept

I am pleased to inform you that your work has now been accepted for publication in PeerJ Computer Science.

Please be advised that you are not permitted to add or remove authors or references post-acceptance, regardless of the reviewers' request(s).

Thank you for submitting your work to this journal. On behalf of the Editors of PeerJ Computer Science, we look forward to your continued contributions to the Journal.

With kind regards,

Reviewer 1 ·

Basic reporting

No Comments

Experimental design

Good

Validity of the findings

No comments

Additional comments

The title of the article is CTCN: A novel credit card fraud detection method
based on CTGAN and TCN.Credit card fraud not only results in financial losses but also damages the reputation of financialTo address the issue of imbalanced data affecting fraud detection performance, we propose a credit
card fraud detection method based on CTGAN and TCN.
institutions, causing people to lose trust in credit card payments. Therefore, credit card fraud detection
has become a crucial task for financial institutions. Here are some comments for the improvement of the manuscript.
1.The Abstract section should be mentioned the experimental findings achieved by the proposed method.
2.The main contribution part is described neatly.
3.Include the motivation of the study.
4.The challenges section is missing.
5.Can you include the flowchart of the algorithm
6.Need to analyses results of more experimental analysis.
7.Check for the grammar and typos correction.

Reviewer 3 ·

Basic reporting

-

Experimental design

-

Validity of the findings

-

Additional comments

The manuscript has been revised as per the reviewer's suggestions and it can be considered for publication.